# GENERATIVE 3D OBJECT PARTICLE DYNAMICS

## ABSTRACT

We introduce ParticleDiffuser, a particle-based 3D trajectory diffusion model that represents scenes as evolving particle graphs, enabling the capture of complex action–object interactions and object deformations. Unlike existing 3D particle dynamics models, which typically rely on deterministic action-conditioned predictors constrained to narrow domains (e.g., individual cloth or soft-body objects), ParticleDiffuser adopts a generative approach trained on large-scale simulated data of deformable and soft objects, and capturing multimodality of future particle trajectories. To support efficient spatiotemporal reasoning, ParticleDiffuser introduces learnable latent vectors that fuse information across particles and employs autoregressive rollouts with latent-variable attention across sequential frame segments, enabling long-horizon 3D video generation. We present two variants: (i) an action-conditioned particle trajectory generator, and (ii) a joint action–object particle trajectory generator. By directly modeling the joint distribution of object particles and actions within a single diffusion process, ParticleDiffuser allows goal-conditioned action generation by steering diffusion toward desired future configurations, eliminating the costly trajectory searches required by traditional MPC methods. Experiments show that ParticleDiffuser generalizes to diverse objects and actions in simulated and real-world settings where deterministic graph-based particle networks quickly fail. It also substantially outperforms MPC baselines in both accuracy and efficiency on manipulation tasks involving a broad spectrum of object types, including rigid and deformable bodies.

## 1 INTRODUCTION

The central premise of world models is to learn predictive models that capture how scenes evolve in response to an agent's actions, modeling both object dynamics and action consequences (Craik, 1943). This raises fundamental questions about scene representation and how temporal structure should be modeled. Some methods predict the future directly in pixel space (Oh et al., 2015), while others adopt physics-inspired representations, modeling the trajectories of 3D scene points (particles) (Li et al., 2018; Shi et al., 2023). In the former, prediction involves generating pixels to reflect object and camera motion. In the latter, the model predicts future scene states by simulating the movement of 3D particles, akin to how physics engines evolve scenes through object-level dynamics.

Recent advances in video generative models have shown that diffusion-based objectives are highly effective for capturing the multimodality of real-world videos (Chai et al., 2023; Zhu et al., 2024). While predicting pixel observations enables scaling to large Internet datasets, it limits the model's physical grounding, often leading to geometric inconsistencies or implausible interactions in generated scenes (Brooks et al., 2024). This raises a key question: can similar advances in generative modeling be extended to **physics-aware** 3D scene prediction?

In this work, we introduce ParticleDiffuser, **a scalable diffusion-based generative model of 3D particle trajectories that captures object dynamics in point-cloud space.** The model recurrently processes point-cloud sequences of arbitrary length, enabling autoregressive rollouts for long-horizon dynamics. To address the high computational cost of all-to-all particle attention, ParticleDiffuser employs a compact set of latent queries that both route attention to particles and summarize past information during recurrence, following ideas similar to scalable video models of Jabri et al. (2022).

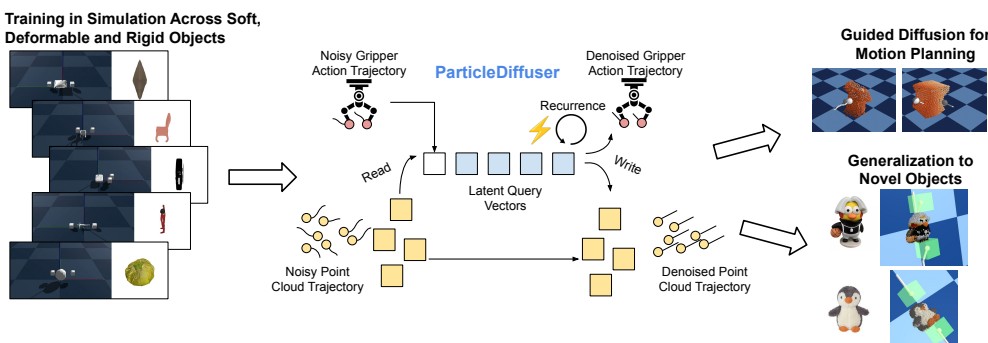

Figure 1: **ParticleDiffuser** is a generative 3D particle trajectory generation model trained from point cloud trajectories of simulated robot interactions with rigid, deformable and soft objects. Thanks to its large scale training and generative objective that can handle multimodality in prediction, it can effectively generalize to novel object shapes and accurately predict their dynamics.

We train ParticleDiffuser on large-scale datasets of 3D point trajectories obtained by simulating interactions between a parallel-jaw robot gripper and diverse objects in the Genesis physics engine (Authors, 2024), covering rigid, soft, and deformable objects. Despite the inevitable sim-to-real gap, the model generalizes effectively to real-world settings, producing physically plausible dynamics across diverse object shapes and actions. This robustness stems from conditioning on point clouds, which transfer more naturally from simulation to reality than pixel-based textures.

We develop two variants of ParticleDiffuser: (i) an action-conditioned object particle trajectory generator, which conditions on 3D fingertip trajectories to generate corresponding object point tracks; and (ii) a joint action-object trajectory generator, which jointly predicts fingertip and object particle trajectories given an initial scene configuration. **By jointly modeling actions and object particles within a single diffusion process, a desired object goal (e.g., location, velocity, or trajectory) provides a gradient signal that steers the denoising process toward action trajectories that achieve the goal.**

We evaluate ParticleDiffuser on synthetic benchmarks and diverse real-world scenes, showing that it achieves lower error drift and greater scalability over long horizons than deterministic graph dynamics models (Shi et al., 2024b). Despite being trained entirely in simulation, it demonstrates strong sim-to-real generalization. We further test the joint-generation variant for robot control using guided diffusion. Compared to random shooting—the standard MPC strategy—guided diffusion in ParticleDiffuser substantially reduces computational cost and accelerates convergence, while maintaining or surpassing task success rates.

To the best of our knowledge, this is the first *generative* 3D particle trajectory model trained across a *broad spectrum of object types*, and the first approach to apply *guided diffusion to planning* while explicitly modeling object interactions in 3D. Existing state-of-the-art particle models are purely deterministic (Zhang et al., 2024), and prior uses of guided diffusion for planning have been limited to modeling the robot body alone (Huang et al., 2025a), without accounting for interactions with external objects.

**Contributions: (1)** A scalable framework that integrates diffusion objectives with 3D particle-based dynamics modeling. **(2)** A curated dataset of 3D point trajectories from diverse robot–object interactions, spanning rigid, soft, and deformable objects in both simulation and the real world. **(3)** A guided diffusion approach for robot–object control that outperforms traditional MPC in both efficiency and success rate. We will release models and code upon acceptance.

## 2 RELATED WORK

**Particle-based dynamics models** Particle based dynamics models represent the environment as a graph of interacting 3D particles. These models operate directly in 3D space, explicitly encoding motion, geometry, and actions through learned interactions. Prior work has used deterministic

graph neural networks to model the evolution of such particle systems via message passing, and applied it in simulating rigid bodies (Li et al., 2018; Battaglia et al., 2016), elastic and plastic materials (Shi et al., 2023; 2024a), fluids (Li et al., 2018; Sanchez-Gonzalez et al., 2020), and granular media (Sanchez-Gonzalez et al., 2020; Wang et al., 2023; Tuomainen et al., 2022). These architectures share structural similarities with transformers employing relative positional encodings (Veličković et al., 2018).

ParticleDiffuser advances this line of work in several key ways. First, **it replaces deterministic prediction with a generative formulation**, enabling the modeling of multi-modal and stochastic behaviors that are common in real-world physical interactions. Second, **it introduces a scalable latent-variable diffusion architecture, approximating expensive full all-to-all particle attention operations** while maintaining expressiveness. Third, unlike previous particle-based models that often train separate graph networks for each material or object type, ParticleDiffuser is **trained across diverse object categories and physical properties**, improving generality and cross-domain applicability. Moreover, ParticleDiffuser allows **for fast and effective control inference through guided diffusion.**

**Planning with diffusion models** A growing body of work has explored planning through guiding diffusion models (Ajay et al., 2022; Janner et al., 2022a; Liang et al., 2023; Du et al., 2023; Yang et al., 2023b; Li et al., 2023; Yang et al., 2023a; Chen et al., 2024; Dong et al., 2023). These approaches train generative models of state and action trajectories and then guide the generation process using a differentiable reward function, either analytically defined to penalize deviation from a goal state (Janner et al., 2022b; Huang et al., 2025b), or learned to predict cumulative rewards or value estimates from sampled trajectories. Recent work (Lu et al., 2025) provides a systematic exploration of the design space for diffusion-based planning. Critically, however, all the aforementioned diffusion-guided planning methods model only the agent's dynamics, such as a robot manipulator or vehicle (Janner et al., 2022b), while assuming the external environment, if at all existent, is static (Huang et al., 2025b). In contrast, ParticleDiffuser is, to the best of our knowledge, the first diffusion-based control framework that jointly models both the robot and the external object it interacts with. This joint modeling allows for guided diffusion to be used to achieve specific object arrangements or placements. Our model generalizes across rigid, soft and deformable objects, which means our framework can handle control across all these scenarios.

# 3 LEARNING GENERATIVE 3D OBJECT PARTICLE DYNAMICS IN SIMULATION

The architecture of ParticleDiffuser is illustrated in Figure 2. It is a diffusion model designed to generate 3D particle trajectories for both an object and its interacting agent. Specifically, the model predicts the 3D displacements of 3D object particles sampled from the object surface as well as the trajectories of the agent's fingertips. By abstracting actions as fingertip particle trajectories, ParticleDiffuser can seamlessly model interactions in an embodiment-agnostic manner, as well as support tools, like sticks applying external forces.

A key advantage of ParticleDiffuser lies in its generative formulation, which is crucial for learning from large-scale datasets of robot-object interactions spanning diverse object types including rigid, soft, and deformable bodies. Unlike deterministic models, which often fail to generalize across such multimodal physical dynamics, ParticleDiffuser is able to model the inherent variability of object behavior. This enables generalization beyond the narrow domains or material-specific setups commonly seen in prior particle-based approaches (Sanchez-Gonzalez et al., 2020; Wang et al., 2023; Tuomainen et al., 2022).

We propose two variants of our model:

1. **Action-Conditioned ParticleDiffuser:** Given an initial 3D point cloud and agent fingertip 3D trajectories, this variant predicts the resulting object particle trajectories. It is well-suited for use in model-predictive control (MPC) frameworks, where it can simulate the effects of sampled action sequences over time.

2. **Joint Action-Object ParticleDiffuser:** Conditioned only on the initial scene point cloud, this model jointly generates both the agent's fingertip trajectories and the corresponding object particle trajectories. This formulation is useful for control via guided diffusion, where a target

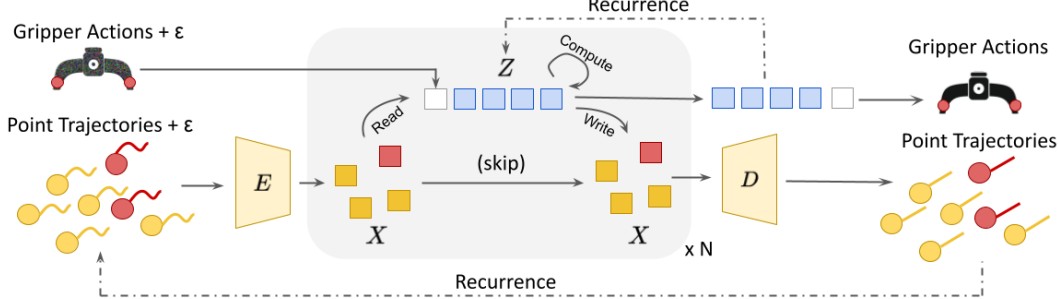

Figure 2: **Architecture of ParticleDiffuser.** 3D point trajectories are encoded into tokens $\mathcal{X}$ and denoised using a diffusion model. The denoising model uses a Read-Compute-Write strategy to keep the bulk of the computation in a lower-dimensional latent space, reducing memory costs, with the action sequence being injected into the latents $\mathcal{Z}$ to integrate action information.

object goal (e.g., desired location or configuration) provides a gradient signal that steers the generation process toward action-object trajectories that satisfy the goal. Unlike MPC, which requires repeated rollouts, guided diffusion enables efficient, one-shot action inference, delivering orders-of-magnitude improvements in computational efficiency.

We provide background on diffusion models in Section 3.1, present the detailed architecture of ParticleDiffuser in Section 3.2, and describe our control method via guided diffusion in Section 3.2.1. Finally, Section 3.3 details the construction of our large-scale dataset of 3D particle trajectories for diverse agent-object interactions.

### 3.1 PRELIMINARIES: DIFFUSION MODELS

Diffusion models consist of a forward (noising) process and a reverse (denoising) process. In the forward process, Gaussian noise is gradually added to a clean data sample $x_0$ over $T$ timesteps, resulting in a sequence of increasingly noisy samples: $q\left(\mathbf{x}_t \mid \mathbf{x}_0\right) = \mathcal{N}\left(\mathbf{x}_t; \sqrt{\bar{\alpha}_t}\mathbf{x}_0, (1 - \bar{\alpha}_t)\mathbf{I}\right)$, where $\mathbf{x}_t$ denotes the noisy version of $\mathbf{x}_0$ at diffusion step $t$, and $\bar{\alpha}_t$ is a pre-defined constant determined by a variance schedule. During training, a timestep $t$ is sampled uniformly, and $\mathbf{x}_t$ is generated using:

$$\mathbf{x}_t = \sqrt{\bar{\alpha}_t}\mathbf{x}_0 + \sqrt{1 - \bar{\alpha}_t}\epsilon_t, \quad \epsilon_t \sim \mathcal{N}(0, \mathbf{I}). \tag{1}$$

The denoising network $\epsilon_\theta$ is trained to minimize the mean squared error between the predicted and true noise: $\mathcal{L} = \left|\epsilon_\theta(\mathbf{x}_t, t) - \epsilon_t\right|^2$. At inference time, the process begins from a Gaussian sample $\mathbf{x}_T \sim \mathcal{N}(0, \mathbf{I})$, and the model iteratively denoises it according to:

$$\mathbf{x}_{t-1} = \frac{\mathbf{x}_t - \sqrt{1 - \alpha_t}, \epsilon_\theta(\mathbf{x}_t, t)}{\sqrt{\alpha_t}} \tag{2}$$

until $t = 0$ is reached and a clean sample $\mathbf{x}_0$ is produced. To accelerate sampling, deterministic alternatives such as DDIM Song et al. (2021) can be used, significantly reducing the number of denoising steps required during inference.

### 3.2 PARTICLEDIFFUSER

**Data Tokenization and Normalization** Our input data consists of 3D point cloud trajectories paired with a sequence of agent action pose changes. Each 3D object trajectory $\tau_o$ is represented as a sequence of $N$ points over $T$ time steps, thus $\tau_o \in \mathbb{R}^{N \times T \times 3}$. We use PointNext layers (Qian et al., 2022) to downsample the spatial dimension of the point cloud and apply a lightweight MLP to compress particle trajectories along the temporal axis. This makes training computationally feasible without noticeably degrading downstream performance. We represent positional information with **rotary positional embeddings** (Su et al., 2020), which encode relative positions in a translation-invariant manner. Compared to absolute encodings such as sinusoidal embeddings, rotary embeddings are better suited for capturing local interactions in 3D space and time, where the center of

the 3D world coordinate frame is arbitrary. We jointly train our spatio-temporal tokenization layers along with the diffusion model. This joint training is necessary due to the lack of large-scale, pre-trained 3D auto-encoders and the limited generalization of existing models to the diverse object interactions and scene dynamics present in our dataset. We apply linear scaling independently to each spatial axis using the 1st and 99th percentiles to normalize the inputs. This quantile-based normalization is more robust to outliers than standard min-max, which is particularly important in dynamic scenes where abrupt contacts or large object motions can introduce extreme values.

**Latent-Based Attention for Memory-Efficient 3D Diffusion**     We adopt **RIN** (Jabri et al., 2022) as the backbone of our 3D diffusion model. RIN is a transformer-based architecture that achieves memory efficiency by replacing all-to-all attentions on input tokens $\mathcal{X}$ with a compact set of learned latent embeddings $\mathcal{Z}$, which mediate the flow of information through a **Read-Compute-Write** process.

- **Read Phase:** Each latent embedding in $\mathcal{Z}$ attends to the input point cloud tokens $\mathcal{X}$, extracting and encoding relevant scene information into the latent space.

- **Compute Phase:** The latents $\mathcal{Z}$ interact with each other via self-attention and feedforward layers, enabling global reasoning and propagation of long-range context.

- **Write Phase:** The input tokens $\mathcal{X}$ attend to the updated latents $\mathcal{Z}$ and update themselves based on the latent-derived context.

This latent-centric attention structure significantly reduces memory usage by shifting expensive self-attention operations from the high-dimensional input space to the smaller latent space.

To effectively integrate action information, we embed the action sequence $a$ using a lightweight MLP. These action embeddings are concatenated with the latents $\mathcal{Z}$ before the Read and Compute phases, allowing the model to condition its predictions on action information while maintaining the memory and efficiency benefits of latent-based computation.

**Recurrent Training and History Conditioning**     To enable long-horizon 3D trajectory generation, our diffusion model must recurrently predict future segments by leveraging information from prior states. Inspired by Jabri et al. (2022), we implement a flexible conditioning mechanism in which the history context is summarized in the latent embeddings $\mathcal{Z}$ of the diffusion model. At each recurrent step, the model generates a short trajectory segment (e.g., several future timesteps) conditioned on a history context window summarizing the past. For **history conditioning**, we retain the latent embeddings from the most recent $\hat{K}$ predicted frames and incorporate them into the current latent set $\mathcal{Z}$. In practice, we find that even a short history window suffices, as the diffusion model is capable of internalizing longer-term dynamics across recurrent steps. Specifically, the model is trained to directly predict in sequences of $8$, and is trained recurrently for $3$ steps. This equates to each training sequence being $24$ timesteps long. We adopt a similar strategy at inference time for long trajectory generation. During training, we simulate the rollout process by recursively feeding the model's own predictions back into its input, rather than relying on ground-truth trajectories. This **recurrent training** approach teaches the model to self-correct and improves its robustness at inference time, when ground-truth sequences are unavailable.

**Output and Losses**     We begin by applying PointNext upsampling layers to restore the output to the original input shape of $N \times T \times 3$. Then, a final linear layer is applied to produce the predicted noise $\hat{\epsilon}$, which is compared to the true added noise $\epsilon$ using an L2 loss during training. More implementation details can be found in Appendix B.

### 3.2.1 PLANNING WITH GUIDED DIFFUSION

Modelling the joint distribution of object particle trajectories and gripper actions within a single diffusion process in our Joint Action-Object ParticleDiffuser variant permits action inference through guided diffusion that steers the generated trajectories towards desired object goal configurations. Specifically, we implement a form of score-based guidance that modifies the reverse diffusion steps using the gradient of a loss function $\mathcal{L}(\mathbf{x}, \mathbf{x}^*)$ that measures deviation of its argument $\mathbf{x}$ from the desired object configuration $\mathbf{x}^*$ (end-location or trajectory for any object particle or subset of them).

We adapt the DDIM denoising process as in He et al. (2023):

$$\mathbf{x}_{0|t} = \mathbf{x}_{0|t} - c_t \nabla_{\mathbf{x}_{0|t}} \mathcal{L}\left(\mathbf{x}_{0|t}; \mathbf{x}^*\right),$$

$$\mathbf{x}_{t-1} = \sqrt{\bar{\alpha}_{t-1}}\mathbf{x}_{0|t} + \sqrt{1 - \bar{\alpha}_{t-1} - \sigma_t^2}\epsilon_\theta\left(x_t, t\right) + \sigma_t \epsilon_t.$$

This process allows us to synthesize action sequences likely to achieve the desired goal, effectively turning the model into a generative policy prior, for *any* goal configuration.

### 3.3 Curating a Dataset of Diverse Robot–Object Interactions in a Physics Engine

Physics simulators are powerful tools for generating large-scale interaction data, offering (1) accurate ground-truth 3D states and action labels, (2) efficient parallelized data generation, and (3) full control over environment parameters, object properties, and agent behaviors. These capabilities make them particularly valuable for training 3D dynamics models, where diverse and precisely labeled spatiotemporal data is essential.

We build our dataset using Genesis (Authors, 2024), a recent simulation engine supporting rigid bodies, articulated objects, and deformable materials. Object meshes and class labels are sampled from Objaverse (Deitke et al., 2023), and GPT-4 (Achiam et al., 2023) is prompted to assign plausible material types and physical properties (e.g., mass, friction, elasticity) based on object class. This produces semantically coherent and physically consistent scenes. We select 3074 meshes from 373 categories.

Interaction sequences are procedurally generated by sampling gripper trajectories that shove, grasp, lift, or drop objects in diverse directions (details in the Appendix). The simulator provides full temporal access to 3D states, from which we extract aligned point-cloud trajectories of both gripper and objects. The gripper is represented by two fingertip key points, with its frame-to-frame pose change serving as the action signal. These paired with evolving object point clouds form the training data for our 3D particle-based diffusion model. We generated 30k videos of 300 to 400 timesteps each in total.

To better match real-world observations, we apply two post-processing steps: (i) keeping only surface points, since internal geometry is not visible in real videos, and (ii) randomly removing subsets of visible surface points around sampled object locations to mimic occlusions and viewpoint limitations. These steps narrow the sim-to-real gap and improve robustness in real-world deployments. More details on the training data generation can be found in Appendix A.

## 4 Experiments

We test ParticleDiffuser on predicting 3D particle trajectories given robot actions, and inferring action sequences that achieve specified object configurations. Our experiments aim to address the following questions: **(1)** How does ParticleDiffuser perform compared to existing deterministic particle dynamics models? **(2)** To what extent does ParticleDiffuser generalize to unseen objects, both in simulation and in real-world data? **(3)** How does planning via guided diffusion in ParticleDiffuser compare against established model-predictive control (MPC) frameworks?

### 4.1 Action-Conditioned 3D Trajectory Prediction

We evaluate ParticleDiffuser and baselines on their ability to predict object particle 3D motion trajectories conditioned on input gripper actions in simulation and the real world.

**Baselines** We compare our model against *Particle Graph Neural Network (GNN)* (Shi et al., 2023; 2024a), a state-of-the-art particle motion prediction model, which represents the object as a graph over object particles where edges capture their spatial proximity. Note that this representation is similar to the one used in ParticleDiffuser with the relative positional encodings, yet, without the latent variable accelerations. *GNN* is a deterministic model and is trained to predict particle displacement trajectories using a regression loss. We train the baseline in our dataset; we represent

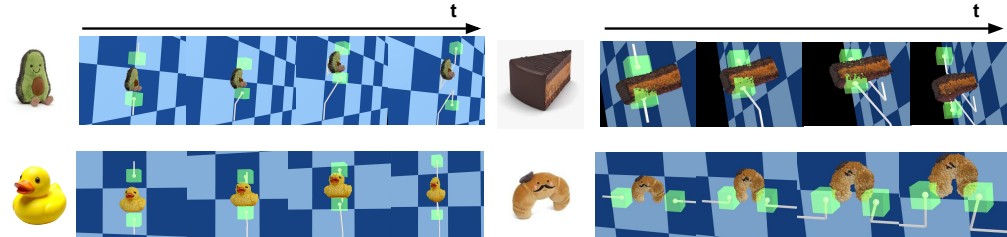

Figure 3: **Qualitative evaluation of action-conditioned trajectory prediction of ParticleDiffuser in the real world.**

the action of the gripper as finger tip nodes which we connect to all object particle nodes. We also consider the following ablative versions of our model: (1) *ParticleDiffuser w/o ActVel* represents the gripper actions with the absolute gripper positions instead of velocities. (2) *ParticleDiffuser w/o Augs* disables our data augmentation strategies, while keeping the rest of the setup unchanged. (3) *ParticleDiffuser Recur2* reduces the number of recurrent steps during training from 3 to 2.

**Datasets** For evaluation in simulation, we consider two object test sets: **(1)** *In-distribution object set*, comprised of the same object instances as the training data. **(2)** *Out-of-distribution object set*, comprised of novel object 3D meshes—not seen during training—to test the model's ability to generalize to novel object categories. We generate 150 videos of 90 timesteps for each of the two evaluation setups. For evaluation in the real world, we extract object 3D point clouds from real images of soft objects. Specifically, in a given image, we segment the object by prompting Gemini 2.5 (Anil & et al, 2023) and reconstruct it into a high-quality 3D mesh using the publicly available method of Hunyuan-3D (Team, 2025). We then sample push and pick-up actions that manipulate the mesh, and use ParticleDiffuser and the baselines to predict the resulting 3D object particle trajectories.

**Evaluation Metrics** For evaluation in simulation, we report the Chamfer Distance (CD), Earth Mover's Distance (EMD), as well as the Mean Squared Error (MSE) between the predicted and ground-truth point clouds. All metrics are averaged over time and across scenes. It is challenging to obtain 3D point trajectory ground-truth in the real world and thus we limit quantitative evaluations to simulation.

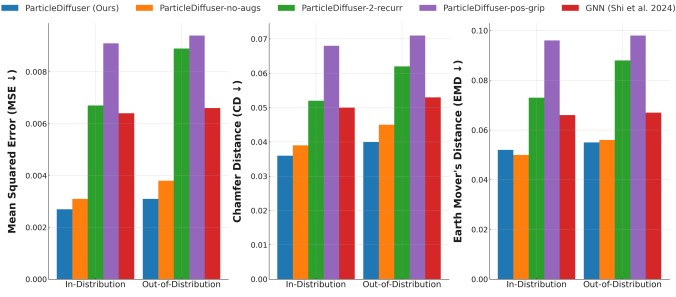

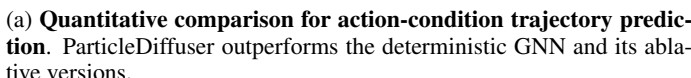

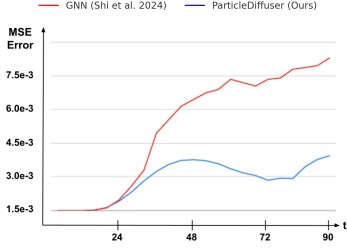

(a) **Quantitative comparison for action-condition trajectory prediction**. ParticleDiffuser outperforms the deterministic GNN and its ablative versions.

(b) **Prediction MSE versus forecasting horizon.** ParticleDiffuser is significantly more accurate than the deterministic GNN baseline at forecasting horizon grows.

Figure 4: **Predicting action-conditioned object dynamics in Simulation**.

We show quantitative trajectory generation results in Figures 4b,4a. ParticleDiffuser outperforms the state-of-the-art GNN models both on in-distribution and out-of-distribution objects. We evaluated the rollout stability of our model and the baseline by measuring the prediction error as a function of rollout length. As shown in Figure 4b, the GNN baseline performs competitively in short horizons,

even slightly outperforming ParticleDiffuser when $t < 24$, which aligns closely with the supervised training range. However, as the prediction horizon increases, the GNN model accumulates compounding errors and gradually drifts away from the true dynamics.

We show qualitative point trajectory generation results of ParticleDiffuser in real world images in Figure 3. ParticleDiffuser effectively generalizes to real-world objects despite being trained in simulation thats to its 3D point cloud representations, its large-scale of training and its generative nature. Additional qualitative results are provided in our website: `https://submit-annonymous.github.io`.

**Ablations**   As shown in Figure 4a, replacing velocity-based gripper control with absolute gripper positions leads to a substantial performance drop, indicating that gripper velocities provide a more informative and stable conditioning signal for modeling object dynamics over time. Disabling data augmentations also reduces performance, particularly in out-of-distribution settings, underscoring their role in enhancing generalization. Limiting training to only two recurrent steps significantly degrades accuracy, highlighting the importance of multi-step temporal modeling for long-horizon predictions. In our experiments, increasing recurrence beyond three steps yielded no further gains. Overall, the full ParticleDiffuser configuration achieves the strongest performance, confirming the necessity of each component.

## 4.2 PLANNING FOR OBJECT MANIPULATION WITH GUIDED DIFFUSION

Next, we evaluate ParticleDiffuser's ability to infer actions to achieve desired object configurations through guided diffusion. Specifically, we use the variant Joint Action-Object ParticleDiffuser with diffusion guidance, as outlined in Section 3.2.1.

**Baselines**   We consider the following baselines: **(1)** *GNN+MPC* (Shi et al., 2023; 2024a), where we combine the graph neural network baseline with MPC for motion planning. **(2)** Action-Conditioned ParticleDiffuser + MPC where we use the action-conditioned variant of our model with MPC.

**Datasets and Evaluation Metrics**   We consider a set of object-gripper manipulation tasks. We randomly sample an initial state and a desired end state. Our model and baselines then infer the action trajectories. We report Mean Square Error between achieved and desired object final configurations.

Quantitative planning results can be found in Table 1 and in Figure 5. Guided diffusion dramatically outperforms MPC in both accuracy and efficiency. In Figure 5 we vary the budget of the MPC search by varying the number of sampled action sequences (3, 5, 10, 20, 50, 100) used for trajectory rollouts within each planning step, and evaluate the corresponding task success. Increasing the planning budget generally leads to improved performance, as the planner explores a broader set of candidate futures and can better optimize for long-term objectives. In Figure 6, we show qualitative results for diffusion guidance. Runtime analysis for our model can be found in Appendix D, and evaluation of our model under a K-best loss can be found in Appendix D.1.

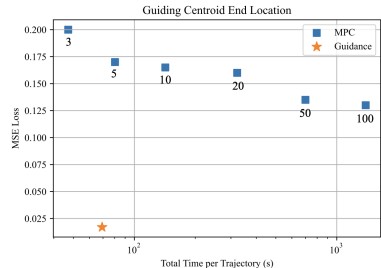

Figure 5: **Guided Diffusion vs. MPC for Planning.**   We vary the search budget of our action-conditioned ParticleDiffuser+MPC and compare with our guided Joint action-object ParticleDiffuser. The guided model is both faster and more accurate than MPC.

## 5 LIMITATIONS AND FUTURE DIRECTIONS

ParticleDiffuser is the first large-scale generative model of object particle dynamics, and the first framework to jointly generate both actions and 3D particle trajectories. Despite these contributions, it has several limitations that point to promising directions for future work. (1) Slow inference: Converting the model to flow-matching or other efficient inference methods could substantially improve

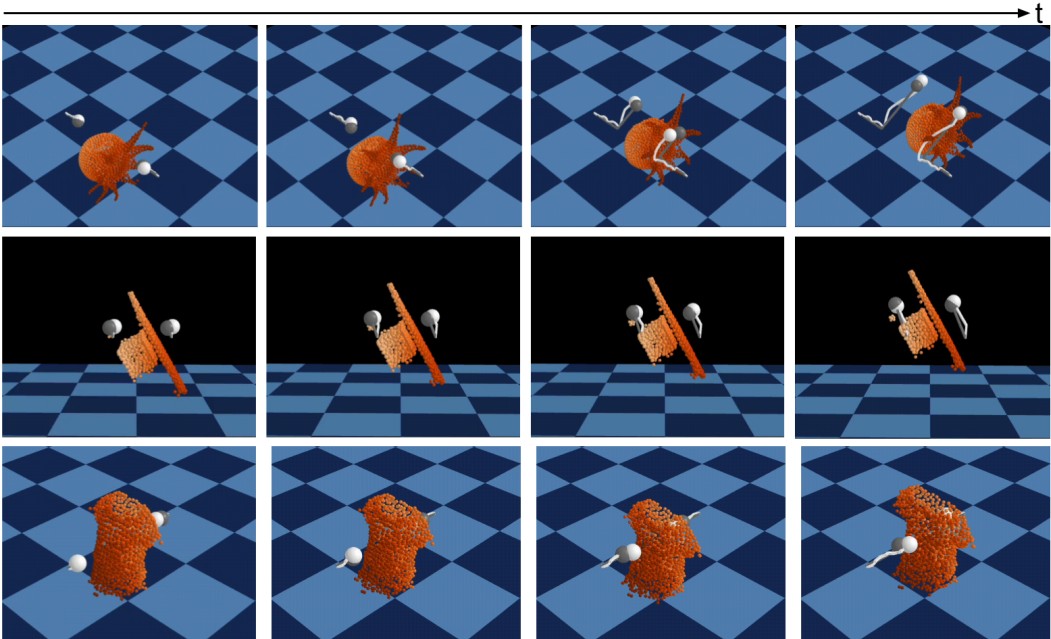

Figure 6: **Motion Planning via Guided Diffusion.** The ground-truth actions are plotted in gray, and the predicted actions are plotted in white. Joint action-object ParticleDiffuser can infer the ground-truth actions that send the object to the desired location.

Table 1: **Planning Evaluations**. We report the Mean Squared Error computed between the predicted and desired object states at the final timestep. We also report the time needed to achieve the reported planning performance.

| Method | MSE (m) ↓ | Planning Time (s) ↓ |
|---|---|---|
| GNN + MPC (Shi et al., 2023; 2024a) | 0.053 | 87.20 |
| Action-Conditioned ParticleDiffuser + MPC | 0.190 | 140.26 |
| Joint Action-Object ParticleDiffuser+ Guided Diffusion | **0.017** | **69.33** |

speed. (2) Limited conditioning: Currently, the model is conditioned only on the initial point-cloud configuration. Incorporating richer conditioning signals—such as textual descriptions of material and physical properties or image-based cues—may help narrow the prediction distribution and yield more accurate dynamics. (3) Single-object training: Our training data includes only single-object interaction sequences. Extending it to cover more diverse action trajectories, multi-object interactions, and larger-scale datasets with millions of samples represents an important next step.

## 6 CONCLUSION

We presented a scalable 3D particle-based generative dynamics model that integrates diffusion objectives with physics-aware modeling. Trained on a diverse set of simulated interactions, our model generalizes to novel object shapes in the real world, producing physically consistent predictions. Unlike traditional deterministic GNN-based particle models, it effectively captures the multimodality of future dynamics and delivers more stable long-horizon rollouts. By jointly modeling actions and object trajectories in point-cloud space, our approach further enables efficient, goal-conditioned planning through guided denoising. We believe this work highlights the potential of combining generative modeling with physics-grounded representations for advancing long-horizon prediction and control, and we hope it will inspire further research on scaling physics-aware generative models for robotics and beyond.

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

## A  DATA GENERATION DETAILS

To populate our simulation scenes, we begin by downloading 3D assets from Objaverse Deitke et al. (2023), sorted by their class labels. From this process, we select 3074 meshes from 373 categories. For each selected object, we merge its geometry into a single watertight mesh and normalize it to a consistent scale and orientation to ensure compatibility with our simulation setup. To assign physically plausible material properties, we query GPT-4 with each object's class label to infer a range of likely values for mass, friction, and elasticity. These ranges reflect semantic priors based on common material characteristics associated with the class. We then randomly sample a specific set of physical parameters within the GPT-inferred range to introduce intra-class diversity while maintaining physical coherence.

Once the object is initialized in the environment, we randomly sample its physical size and starting location within the scene. We also randomly sample a starting position for the gripper. To simulate interactions, we execute a hand-designed manipulation policy that attempts to either push the object or pick it up and drop it. This policy is intentionally designed to produce a mix of successful and failed outcomes by introducing controlled stochasticity. As a result, it generates rich and varied contact dynamics, enabling the model to learn from both effective and unsuccessful manipulation attempts. Please refer to Figure 7 for some examples.

To create evaluation splits, we randomly hold out 10 specific object class labels. During training, we only sample objects that are not part of the held-out classes. At inference time, we evaluate the model on two types of test splits: one using unseen meshes from previously seen classes, and another using meshes from the held-out class labels. This setup allows us to assess the model's ability to generalize to new object instances and novel semantic categories. However, we note that while the held-out class test split restricts direct exposure to those labels, the model may still encounter semantically similar objects during training. More challenging generalization benchmarks, such as excluding broader semantic categories or using adversarially dissimilar classes, are left for future work.

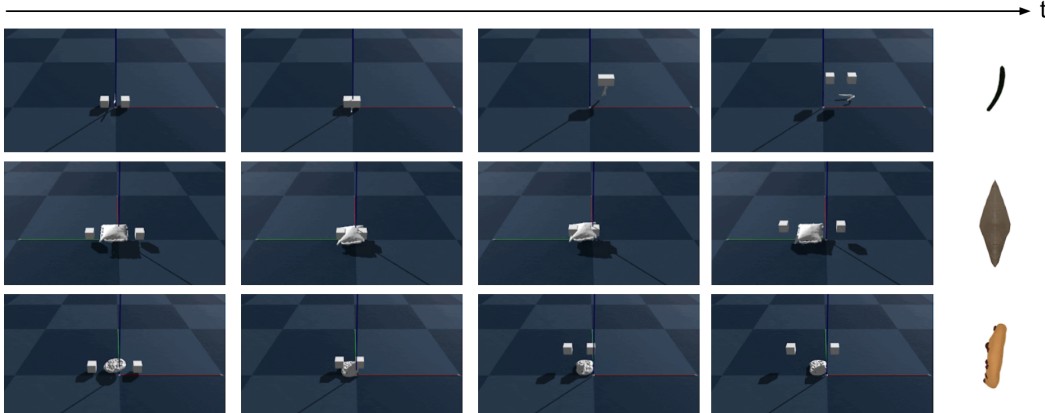

Figure 7: **Simulation Data Generation.** We selected some timesteps in different samples for visualization, with the rendered object mesh on the right.

## B  IMPLEMENTATION DETAILS

We implement our 3D point trajectory diffusion model using a standard DDPM framework, trained to predict noise added to 3D point trajectories over time. The model is composed of a stack of 6 Residual Interaction Network (RIN) blocks, each operating on a latent feature space of 1024 dimensions. The input to the model is a sequence of 3D point clouds, with shape [T, N, 3], where T=24 is the number of time steps and N is the number of points per frame. The model predicts 8 frames at once, and is trained using 3-step recurrence. We use a fixed variance DDPM schedule with 1000 denoising steps for training. For sampling, we use the DDIM sampler with 1000 or 100 inference steps. We train the model using the Lamb optimizer with a learning rate of 3e-4, and apply a cosine learning rate schedule. The model is trained for 200000 total iterations with a batch size of 32. All training is conducted on 8 A100 GPUs, and training takes approximately 36 hours to converge.

## C  ADDITIONAL RELATED WORK: LEARNING WORLD DYNAMICS IN PIXEL AND LATENT SPACE

Learning world dynamics has long been a central objective in robot learning, where the goal is to predict a representation of the future state of the world conditioned on current inputs and agent actions. Dreamer (Hafner et al., 2023) and Daydreamer (Wu et al., 2023) use recurrent state-space models to learn latent transition dynamics through reinforcement learning with reward supervision. In parallel, models such as Genie (Bruce et al., 2024), UniSim (Yang et al., 2023c), and IRASim (Zhu et al., 2024) aim to build fully learned video simulators that can generate plausible future visual rollouts given a sequence of actions and a specific embodiment. In the autonomous driving domain, large-scale models like GAIA-1 (Hu et al., 2023) have demonstrated the ability to generate photo-realistic videos conditioned on past frames, text descriptions, and planned actions.

## D  RUNTIME ANALYSIS

We evaluate the runtime performance and memory efficiency of our model on a system equipped with a single NVIDIA A6000. At inference time, generating a trajectory using DDIM/DDPM with 1000 denoising steps takes 92 seconds per sample, measured over different point trajectories of 24 frames each. The runtime scales approximately linearly with the diffusion steps, hence our approach stands to directly benefit from recent efficient alternatives to diffusion, like Flow Matching.

Table 2: **Model Performance vs. Number of Samples (Best-of-K Selection).** We report the Mean Squared Error using the best sample and the variance of the MSE obtained from rolling out the model multiple times.

| # of Samples | In-Dist | | Out-Dist | |
|---|---|---|---|---|
| | Best MSE | Variance | Best MSE | Variance |
| 1 | 0.0027 | - | 0.0031 | - |
| 3 | 0.0023 | 0.0012 | 0.0030 | 0.0016 |
| 5 | 0.0022 | 0.0011 | 0.0028 | 0.0016 |

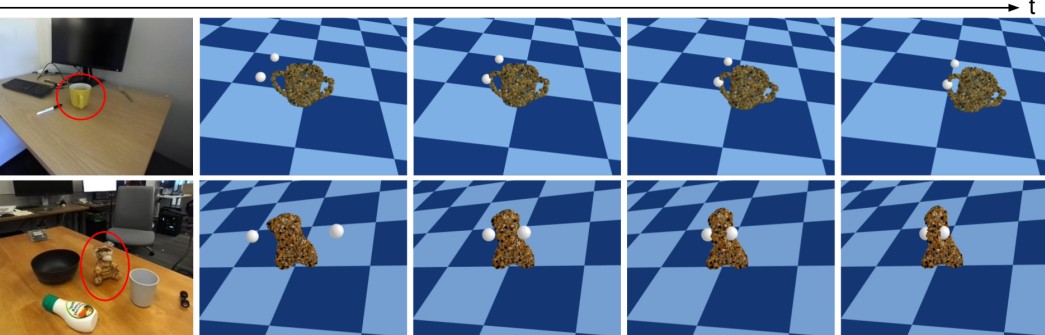

Figure 8: **Qualitative Sim2real Results.** We segment objects in DROID Khazatsky et al. (2024) videos and reconstruct them into complete 3D meshes using foundation VLM Anil & et al (2023) and image-to-3D mesh models Team (2025). We then sample points on the reconstructed mesh and apply gripper actions to them using ParticleDiffuser. We show a cup being pushed in the *top* row and a teddy bear being pinched in the *bottom* row. For more qualitative results, please check our supplementary file.

## D.1 SAMPLE DIVERSITY AND VARIANCE

We analyze the generative diversity of the action-conditioned ParticleDiffuser using best-of-K sampling, selecting the most accurate rollout among multiple stochastic predictions in Table 2. Although this evaluation relies on a ground truth oracle and is not directly deployable, it reveals the model's ability to generate diverse and plausible futures.

As shown in the table, increasing the number of samples improves the best-case MSE, indicating that the model can indeed produce a range of plausible outcomes, among which more accurate ones can be found. This confirms that the model is not simply producing near-identical outputs but is instead generating diverse trajectories. This diversity is especially important for generalization, as it allows downstream components to reason over a richer set of possible futures.

In practice, while ground-truth selection is unavailable at test time, it is possible to integrate our model with downstream selection mechanisms, such as learned value functions, goal conditioning, or planning-in-the-loop, that can harness this diversity in a principled way.

## D.2 DYNAMICS PREDICTION ERROR WITH RESPECT TO PREDICTIVE HORIZON

Our diffusion-based model exhibits significantly greater robustness over long horizons, as shown in Figure 9. Even at $t = 90$, far beyond the training window, it maintains stable predictions and low error. We attribute this to the model's ability to perform denoising across entire trajectories and reason about temporal consistency during sampling, rather than relying solely on autoregressive updates. These results highlight the advantage of our approach in long-horizon forecasting scenarios where input drift poses a major challenge for deterministic or stepwise predictors.

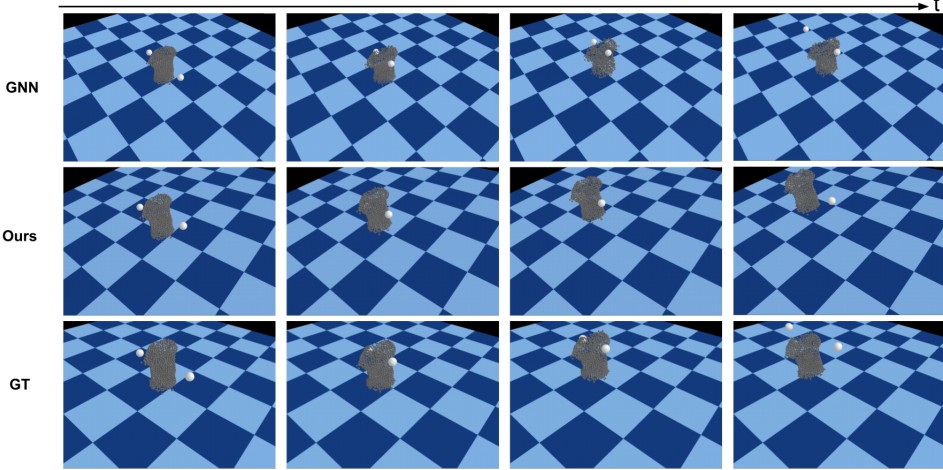

Figure 9: **Long Video Predictions.** ParticleDiffuser is significantly more accurate compared to GNNs for long-term dynamics prediction. Object dynamics predicted by GNNs start drifting and eventually break down and stops moving as the prediction horizon increases. Please check the accompanying mp4 for better visualization effects.

