# OpenReview forum: "Generative 3D Object Particle Dynamics"
_ICLR.cc/2026/Conference — Submitted to ICLR 2026_

### Official Review · Reviewer_gToH · 2025-10-29

**Soundness:** 2
**Presentation:** 2
**Contribution:** 3
**Rating:** 4
**Confidence:** 5

**Summary:**

This paper introduces ParticleDiffuser, a generative 3D particle trajectory model designed to predict the dynamics of object manipulation by capturing the interactions between objects and actions. ParticleDiffuser leverages a diffusion-based approach to simulate the evolution of particles in 3D space, making it applicable to a wide range of object types, including rigid, soft, and deformable objects. To train this model, a dataset of 3D point trajectories is collected by the Genesis simulation.

**Strengths:**

1. ParticleDiffuser presents a novel approach in the generative modeling of 3D particle trajectories, effectively capturing 3D dynamics with action conditioning.
2. A simulation dataset is proposed to train the diffusion model, demonstrating the effectiveness of simulation data on both simulation and real-world cases.

**Weaknesses:**

1. Motivation: The method proposed in this paper uses data from physical simulators for training, which raises concerns about the motivation behind the approach. If the physical simulator can accurately predict the motion of point clouds, why is there a need to train an additional model for prediction?
2. Training Objective: The model is directly supervised with an L2 loss following diffusion training, which brings concerns similar to 2D diffusion models. Specifically, predicting pixel or point observations may not effectively learn physical grounding, as mentioned in lines 44-45 of this paper.
3. Experimental Results: The physical actions generated by this model are relatively simple, particularly in the supplementary video results. Most of the motions are uniform, and some deformations are unrealistic. For instance, the teddy bear in the video does not return to its original shape after being squeezed.

**Questions:**

1. What advantages does the proposed method have over physical simulators like Genesis?
2. Could the model incorporate physical constraints, similar to [1], to improve its accuracy and realism?
3. Could the authors compare their method with physical simulation-based approaches [2, 3]?
4. Is it possible to use point clouds as conditioning to optimize the generation results of video models, similar to [4]?

[1] Inferring Hybrid Neural Fluid Fields from Videos. Advances in Neural Information Processing Systems, 2023, 36: 63595-63608.
[2] PhysGaussian: Physics-Integrated 3D Gaussians for Generative Dynamics. Proceedings of the IEEE/CVF Conference on Computer Vision and Pattern Recognition. 2024: 4389-4398.
[3] DreamPhysics: Learning Physics-Based 3D Dynamics with Video Diffusion Priors. Proceedings of the AAAI Conference on Artificial Intelligence. 2025, 39(4): 3733-3741.
[4] PhysMotion: Physics-Grounded Dynamics from a Single Image. arXiv preprint arXiv:2411.17189, 2024.

---

### Official Review · Reviewer_8ZUo · 2025-10-29

**Soundness:** 3
**Presentation:** 4
**Contribution:** 3
**Rating:** 6
**Confidence:** 3

**Summary:**

This work proposes a diffusion-based dynamics model for control that operates on object surface point clouds and robot gripper contact points named ParticleDiffuser, presenting both action-conditional and non-conditional variants. The model employs several mechanisms to enhance long horizon prediction and computational efficiency including latent attention mechanisms, recurrent training, history conditioning and particle masking augmentations. The authors introduce a novel large-scale simulated object data generation pipeline and evaluate their approach on dynamics prediction and planning in simulation on various objects and many object types, demonstrating improved performance compared to a deterministic baseline and several ablations.

**Strengths:**

**Overview**
- Well written paper.
- The overall method seems novel.
- Interesting use of diffusion over point clouds for dynamics modeling.
- Interesting use of diffusion with recurrence and history propagation to tackle compounding errors in long-horizon prediction.
- Various measures for computational efficiency are integrated in the method, which is particularly important in diffusion-based dynamics models.
- Introduces novel large-scale simulated object data generation pipeline.
- Ablation study highlights some key design choices.
- Evaluation on a large variety of objects and object types.
- Clearly states limitations which facilitate better understanding of the approach as well as highlights avenues for future work.
- Authors intend to publish dataset and code.

I am positive about raising my score if at least some of the concerns raised in the Weaknesses/Questions sections are addressed.

**Weaknesses:**

**Overview**

- Missing implementation details for method and baselines.
- Seemingly inaccurate/misleading claims about real-world evaluation.
- Baselines could be stronger.
- Ablation of the diffusion objective is missing.
- Ablation of the use of particles is missing.
- Experiments consider limited object representation containing only point-cloud information.

**Implementation Details**

Some critical implementation details are missing in my opinion:
1. What is the loss function used for the guided diffusion sampling? What limitation does this choice induce compared to goal-conditioned diffusion?
2. What is a typical size for N, the number of points in the point cloud?
3. How many latent tokens are used? What are the considerations for choosing this hyperparameter?
4. Which algorithm is used to perform MPC and what are the hyperparameter values chosen for the MPC baselines? Specifically, how many planning iterations are used and are they comparable to the number of diffusion iterations?

**Real-world Object Results**

To my understanding, there is no evaluation of the approach in the real world, please correct me if I am mistaken.

This in itself is not a major weakness in my opinion but the claims in the paper are misleading which I believe is a weakness and should be addressed, mainly by clarification and rephrasing. The authors make claims in multiple parts of the paper with relation to real world results such as “the model generalizes effectively to real-world settings” (line 75) and that their method “demonstrates strong sim-to-real generalization” (lines 87-88). Since the proposed method is a dynamics model, real-world should refer to real-world physical dynamics, not just real world images from which object point clouds are extracted. There is actually no sim-to-real here, only real-to-sim. I believe the term real-world in the context of dynamics modeling should be reserved for experiments in real-world dynamics, i.e., with real robot hardware and objects in our physical world. I suggest going over every claim that includes the phrase “real-world” and rephrase it such that this point is clear. The readers should not need to read the paper in depth in order to understand this.

**Baselines and Ablations**

*Deterministic variant of your model*: A baseline/ablation which I believe is clearly missing is a deterministic dynamics model with the exact architecture proposed in your method, i.e., training your architecture with a deterministic dynamics prediction objective instead of the diffusion objective. This will ablate the benefits of diffusion, which are currently not isolated from the architecture. How much of the improved performance is due to the modeling in the latent token space? It is not clear to me that capturing multi-modality is crucial in the tasks you consider. I do not see why the average behavior captured using a GNN/Transformer outputting parameters of a Gaussian distribution can’t perform well in your planning experiments.

*Factored Dynamics Modeling*: The use of particles is treated as a given in the experiment section although the intro suggests there are competing pixel-based paradigms. Do the authors believe there is an appropriate baseline to compare with to demonstrate the efficacy of the particle representation compared to pixels?

**Related Work**

A line of work that I believe is missing in the context of this paper is diffusion-based (unsupervised) object-centric dynamics modeling. Several works have leveraged unsupervised object-centric representations to model dynamics of multiple objects both for video prediction ([SlotDiffusion](https://arxiv.org/abs/2305.11281), [DDLP](https://arxiv.org/abs/2306.05957)) and sequential decision-making ([EC-Diffuser](https://www.arxiv.org/abs/2412.18907)). Specifically, EC-Diffuser is an example of a “diffusion-based control framework that jointly models both the robot and the external object it interacts with” (lines 133-134). Although these methods are not directly comparable to the ones proposed in this paper and have clear distinctions, the literature on factored diffusion is not very large and the above should be mentioned in this context.

**Questions:**

- Why do the authors choose guided diffusion over goal-conditioning?
- Can the authors reference all sections in the Appendix in the main text? In my opinion, the reader should not need to search the Appendix for details. Appendix C is not referenced for example.
- How is the model able to infer physical properties of objects entirely from 3D point clouds? Is there some correlation between geometry and physical properties that the model can leverage for prediction? Is this an assumption your evaluation is based on?
- Can the authors perform an attention analysis on the read/write operations? How much of the latent tokens are actually being used and is there a clear decomposition based on e.g., 3D position or other attributes of the objects? This analysis can shed light on the latent dynamics mechanism and the sources of the improved performance.
- Can the authors include trajectories of the GNN baseline in the real-to-sim objects for comparison? It is hard to evaluate if this behavior is good without any comparison or qualitative metrics.

---

### Official Review · Reviewer_zHbG · 2025-10-30

**Soundness:** 3
**Presentation:** 3
**Contribution:** 3
**Rating:** 6
**Confidence:** 3

**Summary:**

This paper studies learning object-centric physically plausible motion priors. Specifically, it uses particle-based physics simulation and particle trajectories to represent the dynamics. A diffusion model is trained to jointly generate the object particles and the actor motions. The network architecture is designed to be more efficient through learnable cross-attention with more compact tokens, and heavy attention is only applied to these sparse tokens. Because the actor and object motion distributions are jointly modeled, planning and MPC become much easier. Experiments comparing the quality of the prediction and the MPC planning are presented.

**Strengths:**

- This paper studies an interesting and important problem — learning physically plausible motion priors.
- It considers the actuator and the environment jointly and shows planning capability.

**Weaknesses:**

- The method looks interesting, but one important aspect is not verified: how to enable such a model to capture real-world physical properties — not just sim-to-real transfer, but directly learning the distribution from real data.
- From the main comparison Figure 4, only one baseline is considered, which looks a little weak.
- Simulation parameters: the learned distribution is overfitted to the heuristically set parameters of the physical simulator, and the parameters may be augmented (domain randomization). What signal can inform the model to generate the future of one set of parameters instead of another? In other words, if given a few time-step histories, can the model properly realize the physical parameters behind the object and predict the future of this specific configuration?
- Diversity: it is not clear how diverse the synthesized motion is; at least qualitative results must be presented.
- Multiple objects: this is beyond the scope of this project, but it would be interesting to mention or discuss the potential of how to make such a method work with multiple object interactions.

**Questions:**

Please refer to the weakness section. The main questions include: learning from real data, diversity of the generation, system identification capability, and more baselines.

---

### Official Review · Reviewer_jfqk · 2025-10-31

**Soundness:** 2
**Presentation:** 2
**Contribution:** 2
**Rating:** 4
**Confidence:** 5

**Summary:**

This paper introduces ParticleDiffuser, a generative model that learns 3D object dynamics by representing scenes as evolving clouds of particles. The model is trained on a large-scale simulated dataset featuring diverse interactions with rigid, soft, and deformable objects.

**Strengths:**

1. The core strength is the successful application of generative diffusion models to 3D particle dynamics.
2. The paper constructs a dataset of 3D point trajectories from diverse robot–object interactions.
3. The paper proposes a guided diffusion approach for robot–object control, which outperforms traditional MPC in both efficiency and success rate.

**Weaknesses:**

1. The model seems to rely solely on the initial point cloud geometry to predict the future state. Does the model predict the same state when faced with objects of the same category but with different physical properties? If so, this seems inappropriate.
2. Although the method claims to model the dynamics of deformable and soft objects, no significant examples were seen in the demo.
3. To better contextualize the performance of ParticleDiffuser, comparisons with a broader range of methods are necessary. These should include, for instance, using the physics simulator directly for inference or other representative GNN architectures. Furthermore, the description of the GNN baseline is lacking critical details: the specific architecture of the GNN model is not detailed, and the rationale behind the design choice to connect the gripper control nodes to all object particle nodes is not explained.
4. It would be better to incorporate some quantitative metrics in real-world evaluation. If obtaining precise 3D ground-truth is challenging, the authors could consider alternative evaluation methods, such as non-reference metrics and a user study to assess the physical plausibility of generated trajectories.
5. The reported inference time—92 seconds to generate a 24-frame trajectory on an A6000 GPU with 1000 sampling steps—remains prohibitively slow for any real-time application.

**Questions:**

Please see Weaknesses.

---

### Meta-Review · Area_Chair_3PXN · 2025-12-26

**Summary:**

A dynamics model for point cloud data is proposed, based on a diffusion model (and many other technical componenets). I appreciate the technical depth, both in the data generation and the algorithm that integrates many recent techniques.
The results are in simulation, and there are some results with "real" scanned objects, but their dynamics are simulated.

**Reviewer Concerns:**

Reviewers appreciated the technical method.
Major concerns are about:
1. Missing technical information.
2. Misleading claims about real world evaluation.
3. Simulated dynamics too simple
4. Some missing comparisons

Unfortunately, the authors did not provide a rebuttal.

This is a borderline paper. With some more work it could be published. I enocurage the authors to fix the reviewer comments and resubmit.

**Reviewer Scores:**

4,6,6,4.

There is no rebuttal, so hard to say.

---

### Decision · Program_Chairs · 2026-01-26

Reject